# Comparative Evaluation of Three Commercial Elisa Kits Used for the Detection of Aflatoxins B_1_, B_2_, G_1_, and G_2_ in Feedstuffs and Comparison with an HPLC Method

**DOI:** 10.3390/vetsci9030104

**Published:** 2022-02-25

**Authors:** Martha Maggira, Ioannis Sakaridis, Maria Ioannidou, Georgios Samouris

**Affiliations:** Department of Hygiene and Technology of Food of Animal Origin, Veterinary Research Institute, Hellenic Agricultural Organization-DEMETER, Campus of Thermi, 57001 Thessaloniki, Greece; marthamaggira@gmail.com (M.M.); sakaridis@vri.gr (I.S.); ioannidou@vri.gr (M.I.)

**Keywords:** aflatoxins, mycotoxins, ELISA, HPLC, feed

## Abstract

Various analytical techniques for detecting mycotoxins have been developed in order to control their concentration in food and feed. Conventional analytical approaches for mycotoxin identification include thin-layer chromatography (TLC), high-performance liquid chromatography (HPLC), and gas chromatography (GC). Rapid methods for mycotoxin analysis are also becoming increasingly relevant. One of the most common rapid methods for determining these compounds is the enzyme-linked immunosorbent assay (ELISA). The current study aimed to compare three available ELISA kits for the detection and quantification of aflatoxins B_1_, B_2_, G_1_, and G_2_ in spiked feed samples at known quantities. All three ELISA kits were validated and showed good performance with high recovery rates and LOD and LOQ values lower than the MRL. The developed HPLC-FL method was validated for all the compounds determining the accuracy, precision, linearity, decision limit, and detection capability with fairly good results. Unknown feed samples (corn, silage, pellet, barley, wheat, soya, and sunflower) were also tested using the best ELISA kit and HPLC, and the results were compared. Both ELISA and HPLC were proven to be suitable methods for mycotoxin analysis. The analytical technique should be determined primarily by the availability and number of samples.

## 1. Introduction

One-fourth of the world’s crop is considered to be contaminated by mycotoxins during growth or storage, according to a report by the Food and Agriculture Organization of the United Nations (FAO) [1,2]. *Aspergillus*, *Fusarium*, and *Penicillium* are the three main genera of fungi producing mycotoxins [3]. Five major groups of mycotoxins have been described, including aflatoxins, fumonisins, zearalenone, ochratoxin, and deoxynivalenol [4]. Among these groups of mycotoxins, aflatoxins (AFs) are highly toxic and are known to contaminate a wide variety of foods and feeds such as maize, peanuts, cereals, groundnuts, dried fruits, meat, and milk-based products [5,6,7].

Aflatoxins are genotoxic carcinogens that are mainly synthesized by two fungal species: *Aspergillus flavus* and *Aspergillus parasiticus*. These fungal species usually grow in tropical and subtropical regions where the climate is warm and humid [8,9]. The four most important aflatoxins are aflatoxin B_1_, B_2_, G_1_, and G_2_, also called AFB1, AFB2, AFGl, and AFG2. Among them, AFB1 is considered to be the most toxic. Aflatoxins cannot be detected by smell or taste; they fluoresce in ultraviolet light and are resistant to temperatures above 320 °C without decomposing [10]. The consumption of contaminated food and feed by humans and animals could cause serious threats to their health, including hepatotoxicity, teratogenicity, and immunotoxicity [11,12]. Therefore, aflatoxins pose a particular health risk for humans and animals, and consequently, the European Union has set maximum permissible limits for the sum of aflatoxins B_1_, B_2_, G_1_, and G_2_ in all cereals and their derivatives, including processed cereal products, which are set at 4 μg/kg, and for aflatoxin B1, being by far the most toxic compound, content alone is set at 2 μg/kg [13]. Since our samples were mainly crops of cereals, we used those limits and not the ones concerning mixtures of feedstuffs.

So far, the methods used for the detections of aflatoxins in feedstuffs include chromatographic, spectroscopic, and immunochemical methods. The most common chromatographic methods are thin-layer chromatography (TLC), high-performance liquid chromatography (HPLC), and gas chromatography (GC). The chromatographic techniques are very sensitive, but they require highly trained technicians and expensive equipment [14]. Furthermore, most chromatographic methods require solid-phase pre-treatment and immunoaffinity techniques in order to reduce interferences and increase their efficiency in detection [15]. Despite their flaws, they were widely used, and HPLC is considered as the reference method for the qualitative and quantitative determination of mycotoxins [16,17]. However, because of the complicated and time-demanding sample preparation that is required in HPLC, immunological methods like ELISA are preferred in research studies and in routine analyses [18,19]. ELISA has the advantage of being a cheap, rapid method, with low sample volumes and relatively fewer preparation procedures than other methods, which at the same time has high specificity and sensitivity and fast responses with high repetition capabilities [20,21]. However, the accuracy of ELISA can sometimes be influenced by the nature of the mycotoxin, the sample preparation process, and the nature of the material, although its accuracy and reproducibility can be improved by having a previous separation procedure [22]. Furthermore, the “matrix interference” that sometimes appears while performing ELISA can lead to false positive or negative results by over valuating or devaluating the concentration of aflatoxins in the samples [23].

Our objective was to compare three commercial ELISA kits for the detection and quantification of aflatoxins B_1_, B_2_, G_1_, and G_2_ in spiked feed samples at known concentrations. The ELISA kit that was proven to be the most efficient was then compared with an HPLC technique in terms of its efficacy and reliability. The testing of unknown feed samples (corn, silage, pellet, barley, wheat, soya, and sunflower) was also performed with both analytical methods along with a comparative evaluation of the results.

## 2. Materials and Methods

### 2.1. ELISA

#### 2.1.1. ELISA Kits and Instruments Used

Three commercial ELISA kits were selected for the detection of the aflatoxins B_1_, B_2_, G_1_, and G_2_ in feed: (A) ELISA AgraQuant Total Aflatoxin 1/20, Romer Labs Singapore Pte Ltd. (Jalan Bukit Merah, Singapore); (B) ELISA BIO SHIELD M1 ES, ProGnosis Biotech A.E. (Larissa, Greece); and (C) ELISA RIDASCREEN Total Aflatoxin, r-biopharm (Darmstadt, Germany). An ELISA reader TECAN infinite F50 (Grödig, Austria) was used for the photometrical evaluation of the results. Eppendorf Research plus certified pipettes were also used for the preparation of standard solutions and spiking of feed samples at the appropriate concentrations.

#### 2.1.2. Chemicals and Reagents

Standard solution of AFG1 (2.07 μg · mL^−1^ in acetonitrile), AFG2 (0.49 μg · mL^−1^ in acetonitrile), AFB1 (1.99 μg · mL^−1^ in acetonitrile), and AFB2 (0.49 μg · mL^−1^ in acetonitrile) were obtained from Apollo Scientific LTD (Cheshire, UK). HPLC-grade acetonitrile (ACN), methanol (MeOH), and water were purchased from Sigma-Aldrich (Steinheim, Germany). Feed samples were obtained from individual in northern Greece.

#### 2.1.3. Sample Preparation

A mixture of durum wheat and barley was used as a feed sample in the comparative evaluation of commercial ELISA packages. The sample was initially examined by high-performance liquid chromatography (HPLC) for the presence of Aflatoxins B_1_, B_2_, G_1_, and G_2_ (AFB1, AFB2, AFG1, and AFG2, respectively) with a clear background signal. The sample was sealed airtight and kept at −20 °C until the day of analysis.

The sample was transferred to room temperature for thawing a few hours before the analysis. This was followed by the spiking of feed samples at known AFB1, AFB2, AFG1, and AFG2 concentrations. The maximum residue limit (MRL), i.e., 4 μg · kg^−1^ for all AFs, was defined as the analysis area.

The same sample preparation protocol was implemented for all ELISA kits according to the manufacturer’s instructions. Firstly, the spiking of 20 g of a representative sample was at known concentrations of aflatoxins. This was followed by the adding of 70% MeOH until 100 mL and mixing in a blender for 10 min. The mixture was then vacuum filtered through a Buchner funnel with Whatman # 1 filter paper, and the extract was taken to a clean beaker. Prior to analysis, the extract was subjected to pH measurements for AgraQuant 1/20 and BioShield ES Total AFs, while for Ridascreen Total AFs, 100 μL of the filtrate was diluted with 600 μL of distilled water.

#### 2.1.4. Standard Solution Preparation

The solvent used to prepare the standard solutions was a 1:1 methanol–water solution. Initially, concentrations of 50 μg · L^−1^ for AFG2 and AFB2 and 100 μg · L^−1^ for AFB1 and AFG1 were prepared for the four aflatoxins under study. Mixture solutions were then prepared at various concentrations to plot the reference curves and validate the ELISA kits. The ratio of aflatoxins B_1_, B_2_, G_1_, and G_2_ was 3:1:1:1, respectively, and was based on the MRLs of the compounds as defined in the current legislation (MRL AFB1 = 2 μg · kg^−1^ and MRL Total AFs = 4 μg · kg^−1^). The concentrates were stored at −20 °C for the entire validation period of the ELISA kits while the working standards were prepared daily. 

The analyses were performed according to the instructions of each kit manufacturer. All samples remained completely protected from light during the experiment to prevent aflatoxins degradation.

### 2.2. HPLC

#### 2.2.1. HPLC-FL Instrumentation

The Perkin Elmer Series 200 HPLC system was utilized for the chromatographic determination of aflatoxins residues in feed. It was outfitted with a Fluorescence Detector (FL) and a 100 μL loop (Perkin-Elmer, Shelton, CT, USA). The mobile phase was degassed using a Perkin Elmer Series 200 vacuum degasser that was directly plugged into the solvent reservoir. For the separation of the investigated analytes, a MYCOTOX C18 5 μm, 4.6 × 250 mm analytical column obtained from Pickering Laboratories (Mountain View, CA, USA) was utilized. Online photochemical derivatization was carried out with the use of a commercially available system, the UVE 154 LCTech GmbH (Dorfen, Germany), which was installed between the separation column and the fluorescence detector. The evaluation software was Total Chrom V6.2.0.0.1 with LC instrument control Perkin Elmer.

A glass vacuum filtration apparatus obtained from Alltech Associates (Deerfield, IL, USA) was employed for the filtration of the solvents using cellulose nitrate 0.45 μm membrane filters from Sartorius Stedim Biotech GmbH (Goettingen, Germany). For the processing of feed samples, a Vortex Genie 2 (Bohemia, NY, USA), an ultrasonic bath AM-9 Aquasonic Cleaners (Sherwood, AR, USA), and a Hettich Universal centrifugation (Tuttlingen, Germany) were employed, while immunoaffinity columns (Afla Total) were used for the isolation of the aflatoxins (Vicam, Milford, MA, USA). Fioroni 13 mm × 0.2 mm microfilters were used to filter the samples (Ingre, France). Furthermore, the standard solutions were prepared using a 20–200 μL micropipette Eppendorf Research (Hamburg, Germany).

#### 2.2.2. Chromatography

Isocratic elution was used to separate target analytes using A: MeOH, B: ACN, and C: water. The volume ratio was 22:22:56 to A–B–C. The flow rate was set at 1 mL min^−1^ and the time of analysis was 16 min. The analytical column was operated at room temperature, and the analytes were monitored at 365 and 430 nm by an FL detector.

#### 2.2.3. Sample Preparation Prior to HPLC-FL Analysis

A mixture of durum wheat and barley was used as a feed sample for the validation of the HPLC method. Firstly, 50 g of feed and 5 g of NaCl were blended for 1 min with 200 mL of MeOH–H_2_O (80:20). The extract was filtered and collected in a clean vessel, and 10 mL of the filtered extract was diluted with 40 mL of HPLC water and filtered. Then, 10 mL of the diluted extract was passed through the AflaTotal affinity column, with the aid of a plastic syringe barrel, at a stable rate of 1–2 drops/second until air came through the column. After removing the plastic syringe barrel, the column headspace was filled with water. Using a new syringe barrel, the AflaTotal affinity column was washed with 10 mL of HPLC water at a rate of 2–3 drops/second. This procedure was carried out twice. Elution was accomplished by passing 1 mL MeOH through the column and then adding 1 mL purified water to the eluent. The eluent was gathered and filtered, and 100 L aliquots of the obtained samples were injected into the HPLC apparatus.

## 3. Results and Discussion

### 3.1. Validation of ELISA Kits

#### 3.1.1. Linearity Equations of Reference Curves

The plotting of the reference curves was based on the triplicate analysis of the standard solutions supplied in each commercial package (Table 1). According to the instructions of each manufacturer, the averages of the absorption values of the standard solutions (B) were first calculated, then divided by the absorption value of the standard zero solution (standard solution 0 or zero standard) (B0) and multiplied by 100. The plotting of the standard curves and their equations (Figure 1) was based on the absorption values (Y-axis) versus the concentrations of the standard solutions (logarithm X), while the value pairs of at least four models (absorption–concentration) were used as the linearity region. From the resulting equations, the working range, the coefficient correlation, the slope, and the point of intersection with the Y-axis (intercept) were determined. The resulting equations were later used to calculate the concentrations of aflatoxins B_1_, B_2_, G_1_, and G_2_ in the spiked samples. Only in the case of Ridascreen Total AFs, the concentration calculated from the reference curve in ng kg^−1^ was multiplied by a factor of 35 and then divided by 1000 (μg · kg^−1^). The comparative evaluation of the three commercial ELISA kits was carried out using those calculated concentrations.

#### 3.1.2. Accuracy and Repeatability (Within-Day Repeatability)

For the repeatability test, three feed mixture samples were spiked with standard working solutions of aflatoxins B_1_, B_2_, G_1_, and G_2_ at each concentration of 2, 4, and 6 μg · kg^−1^. Each sample was analyzed three times and from the results obtained, the mean concentration, the standard deviation, the % relative standard deviation, and the recovery were calculated. The experimental results after their statistical processing are presented in Table 2.

#### 3.1.3. Limit of Detection (LOD) and Limit of Quantification (LOQ)

For the calculation of LOD and LOQ, 20 blank samples of feed mixture were analyzed, and then the average of the measured concentration and the standard deviation were calculated. The calculations were based on the following relationships:LOD = average measured concentration + 3 × SDLOQ = average measured concentration + 10 × SD

This was followed by the statistical analysis of the results, which are presented in Table 2.

All ELISA kits showed satisfactory accuracy and repeatability at concentrations of 2 and 4 μg · kg^−1^ when analyzing the spiked feed samples three times in the same day (within-day repeatability). However, only the kits of the companies “Agraquant” and “BioShield ES” gave satisfactory recovery values (95.00% and 81.11%, respectively) at the concentration of 6 μg · kg^−1^, while the kit of the company "Ridascreen" showed a clear decrease in the recovery values (R: 61.41%) when analyzing feed samples of higher concentration (6 μg · kg^−1^). 

Regarding the LOD parameter, all kits showed values much lower than the MRL, ranging from 1.04 to 1.59 μg · kg^−1^. The kit of the company “BioShield ES” had the lowest detection limit at the concentration of 1.04 μg · kg^−1^, followed by the kits of the companies “Ridascreen” and “Agraquant” with detection limits of 1.40 and 1.59 μg · kg^−1^, respectively (Table 2). 

As far as the limit of quantification was concerned, the kit of the company “BioShield ES” gave lower values (1.80 μg · kg^−1^) compared to the other two kits of the companies, “Agraquant” and “Ridascreen” (2.47 and 3.15 μg · kg^−1^, respectively) (Table 2). For the above-mentioned reasons, the kit of the company “BioShield ES” was selected for the application in unknown feed samples.

### 3.2. Chromatography

Isocratic elution was used to separate the target analytes. A typical chromatogram is presented in Figure 2. The retention times were calculated at 9.2 min for AFG2, 10.4 min for AFG1, 11.3 min for AFB2, and 13.6 min for AFB1.

The protocol described in the Materials and Methods was applied for the sample preparation. The sample preparation process provided is straightforward, uses few organic solvents, and produces a clear background signal. Furthermore, the suggested approach has the benefit of using online photochemical derivatization. Several methods of derivatization are available, such as pre-column treatment with trifluoroacetic acid (TFA) [24] and post-column derivatization with iodine [25,26]. However, these procedures are time-consuming and have a number of drawbacks, including the use of hazardous chemicals, the instability of the derivatives, and low day-to-day repeatability. Photochemical derivatization is reagentless, fast, and simple to use, resulting in great sensitivity and reproducibility. The chromatograms of a blank and a spiked sample are depicted in Figure 3a,b, respectively.

#### HPLC-FL Method Validation

The developed HPLC-FL method was validated in terms of sensitivity, linearity, decision limit (CCα), detection capability (CCβ), accuracy, and precision according to European Decision 200/657/EU [27]. The validation was performed using the total MRL level. Blank samples were feed samples that had been analyzed and found to have no detectable residues of the analytes. 

Standard solutions showed linearity for all target analytes within the range of 0.05 to 10 μg · kg^−1^ and showed high correlation coefficients (0.999–1.000). Moreover, calibration curves were constructed using spiked feed samples after sample preparation, samples within the range of 2–8 μg · kg^−1^ total concentration. 

Limits of detection (LOD) and quantification (LOQ) were estimated as the concentration giving a signal-to-noise ratio of 3:1 and 10:1, respectively. The calculated LODs for all analytes were found to be lower than the respective MRL values. The good resolution between the chromatographic peaks of analytes and the absence of interferences in the spiked feed samples indicate that a good selectivity was achieved. 

The precision of the method was estimated by taking into account the within-day repeatability and between-day precision. The within-day repeatability was determined by consecutive (*n* = 3) measurements of three spiked samples at concentration levels of 1 μg · kg^−1^, 2 μg · kg^−1^, and 3 μg · kg^−1^ for AFB1 and 0,3 μg · kg^−1^, 0,6 μg · kg^−1^, and 1 μg · kg^−1^ for the rest of the compounds. These concentration levels represent the 0.5 × MRL, MRL, and 1.5 × MRL of the AFB1 and 0.5 × MRL, MRL, and 1.5 × MRL of the total compounds, respectively. By comparing extracted compound peak area ratios to values determined from spiked calibration curves, the relative recovery rates of spiked samples were assessed. The method’s between-day precision was determined by doing triplicate analyses at the same concentration levels over three days. Over a three-day period, each concentration was determined in triplicate.

After spiking 20 blank feed samples at MRL of the total AFs, the decision limit (CCα) was determined. The capability of detection (CCβ) for the same blank samples was also estimated at the corresponding CCα level of the total analytes. Table 3 displays the CCα and CCβ values, as well as the values produced from the validation procedure for the evaluated parameters.

There were some variations observed after the validation of the two procedures when comparing ELISA and HPLC. The intra-day recovery percentage ranged from 88 to 110%, with RSD values lower than 9.3%. Similar mean recoveries were also reported by other studies [28,29]. The recovery rates for the ELISA kit that was selected were from 72 to 97% (Table 2), higher than recovery rates reported in previous studies [30]. Despite being less reliable in previous decades, ELISA has improved significantly in recent years and is now competitive with other analytical techniques, a finding that our study also verified. ELISA, in particular, revealed higher absolute recovery rates and slightly improved precision, indicating its potential for aflatoxin screening in feed samples [31]. However, for the target compounds, HPLC demonstrated improved linearity and lower LOD and LOQ values. Another advantage of the HPLC-FL analysis is its capability to detect AFB1, AFB2, AFG1, and AFG2, in contrast with ELISA that can determine only the total concentration of the aflatoxins. The ELISA approach has some advantages, including less time required and simplicity, while HPLC, like other chromatographic procedures, is more sensitive and specific than ELISA. The availability and number of samples affect the selection of analytical techniques.

### 3.3. Application in Unknown Feed Samples

Thirty samples of animal feed were tested for the presence of AFs residues with both methods. In more detail, the samples analyzed were 11 corn, 6 corn silage, 4 pellet, 2 barley, 2 wheat, 2 soya silage, 1 soya, 1 sunflower, and 1 mix sample. As mentioned before, they were collected from individual farms in northern Greece.

Firstly, the samples were analyzed by the ELISA kit in duplicate according to the manufacturer’s instructions. During the monitoring of the positive feed samples, the ELISA kit determined AFs in 12 out of 30 samples (40%); however, only two samples had AFs at concentrations greater than the allowable limit (0.06%). Following that, feed samples were prepared using the sample preparation procedure given in Materials and Methods before being analyzed using the HPLC-FL method presented herein. With certain exceptions, the HPLC method appeared to be congruent with the ELISA results. Specifically, AFs were determined in 9 out of 30 samples (30%), and one sample contained AF above the MRL (0.04%) (Table 4). The values of mycotoxins from the samples that do not agree with the two methods seemed to be slightly higher in ELISA in comparison to values obtained by HPLC, which is consistent with the results of earlier studies [32,33]. This disagreement with the ELISA method is assumed to be related to the latter’s overestimation. The results obtained with ELISA in comparison to those obtained with HPLC for the samples that were found to be above the LOQ are given in Table 4.

## 4. Conclusions

In conclusion, this study validates three different ELISA kits and HPLC method for detecting and quantifying mycotoxins in feed samples. 

In terms of the comparative evaluation of the ELISA kits, the kits of the companies “Agraquant” and “BioShield ES” showed high recoveries for the AFs. When it came to the LODs and LOQs, all kits had much lower values than the MRL. The kit with the slightly better performance (satisfactory accuracy and repeatability and the lowest LOD and LOQ) was chosen for the screening of unknown feed samples.

Moreover, the current study demonstrates that the developed HPLC-FL method was found to be accurate, sensitive, and repeatable, and it is suggested for determining AFB1, AFB2, AFG1, and AFG2 in feed samples. After the appropriate sample preparation, the analytes were successfully separated from the complex substrate of animal feed and yielded satisfactory recovery rates. Furthermore, the use of the photochemical reactor in the current study was found to make the derivatization step a reagentless, fast, and easy procedure. Both ELISA and HPLC were shown to be suitable methods for mycotoxin analysis. The choice of analytical procedure should primarily be determined by the availability and number of samples.

## Figures and Tables

**Figure 1 vetsci-09-00104-f001:**
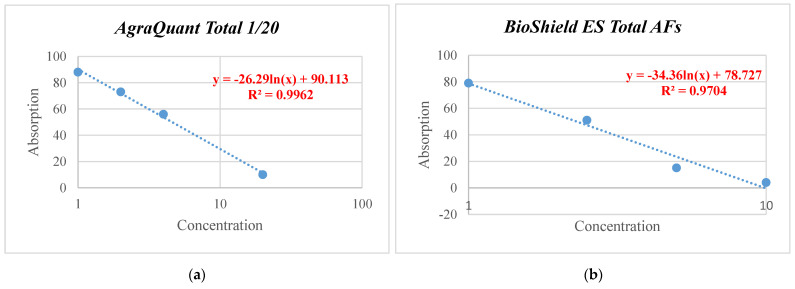
Reference curves of the three (ELISA) kits AgraQuant Total (**a**), BioShield ES Total (**b**) and Ridascreen Total (**c**) where the straight lines are linear regressions of the absorbance values.

**Figure 2 vetsci-09-00104-f002:**
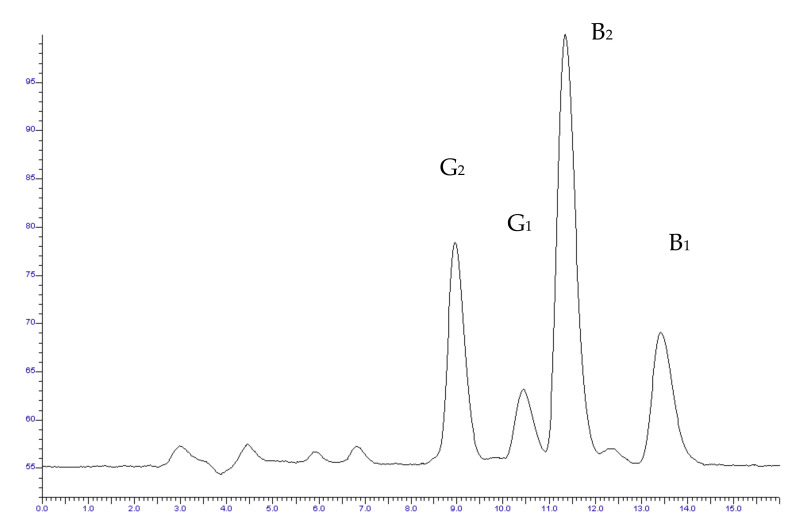
Typical chromatogram of standard solution of examined aflatoxins (G_2_, G_1_, B_2_, B_1_) at the concentration of 1 μg · L^−1^.

**Figure 3 vetsci-09-00104-f003:**
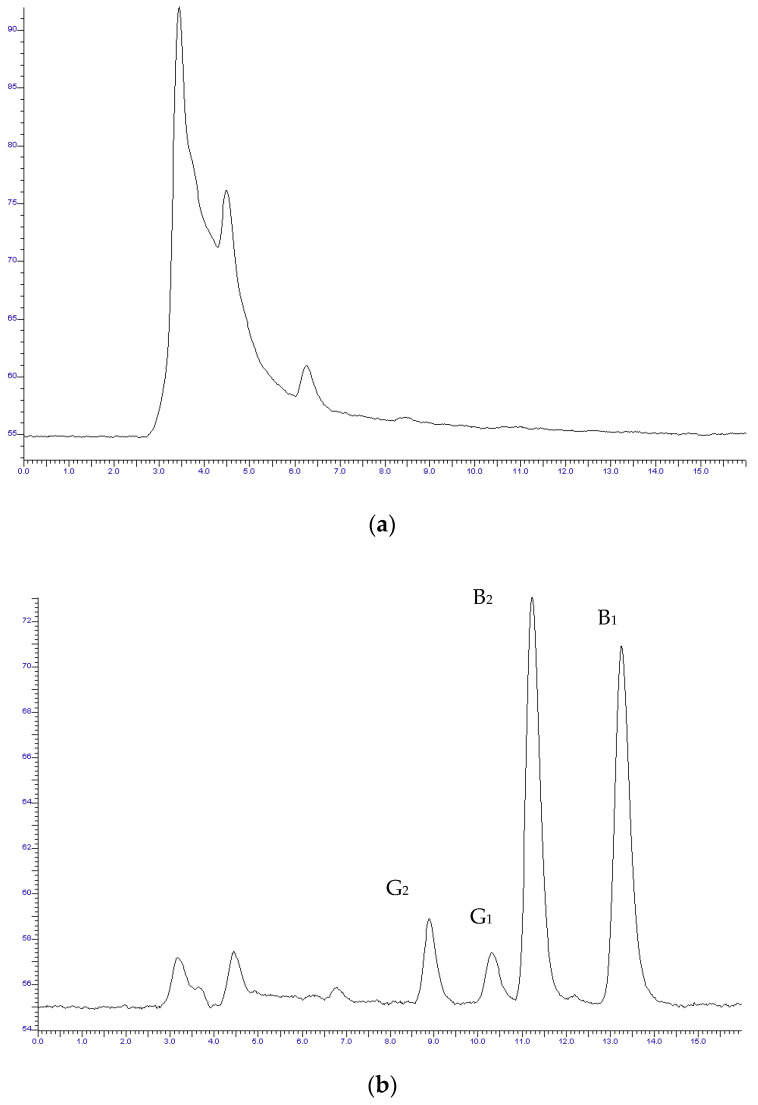
Chromatogram of (**a**) blank feed sample and (**b**) spiked sample at a concentration of 4 μg · kg^−1^ for all the examined compounds (G_2_, G_1_, B_2_, B_1_).

**Table 1 vetsci-09-00104-t001:** Concentrations of standard solutions of the three commercial ELISA kits.

Standard Solutions	Agraquant Total 1/20 (μg · kg^−1^)	BioShield ES Total Afs (μg · kg^−1^)	Ridascreen Total Afs (ng · kg^−1^)
Std1	0	0	0
Std2	1	1	0.05
Std3	2	2.5	0.15
Std4	4	5	0.45
Std5	10	10	1.35
Std6	20		4.05

**Table 2 vetsci-09-00104-t002:** Results from the comparative evaluation of the three commercial ELISA kits for the detection and quantification of AFB1, AFB2, AFG1, and AFG2 in feed.

ELISA Kits	Within-Day Repeatability (*n = 9*)				LOD (μg · kg^−1^) (Xmean + 3 × SD) *n_blank_* = 20	LOQ (μg · kg^−1^) (Xmean + 10 × SD) *n_blank_* = 20
	Total Aflatoxin True Concentration (μg · kg^−1^)	Accuracy (Xmean ± SD)	Precision (%RSD)	Recovery (%R)		
AgraQuant Total Aflatoxin Assay 1/20	2	1.59 ± 0.11	7.20%	79.64%	1.59	2.47
	4	3.80 ± 0.41	10.68%	95.00%		
	6	6.10 ± 0.45	7.34%	101.66%		
BioShield Total ES	2	1.45 ± 0.11	7.43%	72.51%	1.04	1.80
	4	3.24 ± 0.12	3.70%	81.11%		
	6	5.79 ± 0.23	4.05%	96.55%		
Ridascreen Aflatoxin Total	2	1.87 ± 0.20	10.56%	93.45%	1.40	3.15
	4	2.84 ± 0.74	26.26%	70.91%		
	6	3.68 ± 1.09	29.50%	61.41%		

LOD: Limit of Detection, LOQ: Limit of Quantification, SD: Standard Deviation.

**Table 3 vetsci-09-00104-t003:** Validation parameters for the determination of AFB1, AFB2, AFG1, and AFG2 in feed.

Compound	Linearity R^2^	LOD (μg · kg^−1^)	Intra-Day Recovery (%) RSD (%)	Inter-Day Recovery (%) RSD (%)	CCα (μg · kg^−1^)	CCβ (μg · kg^−1^)	Error α	Error β	MRL (μg · kg^−1^)
AFB1	0.999	0.19	98.9–103.7 <9.1	91.9–101.0 <9.3	-	-	-	-	2
AFB2	1.000	0.18	97.6–102.6 <6.5	79.4–105.4 <23.3	-	-	-	-	-
AFG1	0.999	0.13	88.3–110.3 <5.0	82.3–118.0 <16.5	-	-	-	-	-
AFG2	0.999	0.16	97.7–102.0 <9.3	86.0–123.9 <16.1	-	-	-	-	-
Aflatoxins	-	0.61	-	-	4.67	4.98	0.55	0.30	4

LOD: Limit of Detection, RSD: Relative Standard Deviation, CCα: Decision limit, CCβ: Detection capability, MRL: Maximum Residue Limit.

**Table 4 vetsci-09-00104-t004:** The results from the analysis of the samples in which AFs values were above LOQ as determined by ELISA and HPLC-FL method.

Number of Sample	Type of Feed	ELISA Method (TOTAL μg · kg^−1^ ± SD)	HPLC-FL Method (TOTAL μg · kg^−1^ ± SD)
1	Soya	2.11 ± 1.10	1.03 ± 0.78
4	Corn	2.10 ± 0.91	0.92 ± 0.82
7	Corn	2.02 ± 1.64	0.92 ± 0.81
8	Corn	2.99 ± 1.72	2.63 ± 1.02
11	Pellet	2.46 ± 1.21	1.86 ± 0.97
12	Corn	1.85 ± 0.98	1.35 ± 0.88
15	Corn	5.49 ± 2.29	3.05 ± 1.08
16	Corn	3.12 ± 1.41	2.42 ± 0.92
17	Wheat	2.48 ± 1.08	1.23 ± 0.58
19	Corn silage	6.21 ± 1.31	4.52 ± 1.18
22	Corn silage	2.49 ± 0.95	1.21 ± 0.52
25	Corn	1.48 ± 0.89	0.98 ± 0.32

SD: Standard Deviation.

## Data Availability

The data presented in this study are available on request from the corresponding author.

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
