# Peer review of "Comparative Evaluation of Three Commercial Elisa Kits Used for the Detection of Aflatoxins B1, B2, G1, and G2 in Feedstuffs and Comparison with an HPLC Method"

_vetsci, 2022, doi:10.3390/vetsci9030104_

Round 1
Reviewer 1 Report
GENERAL COMMENTS
- Provide additional relevant references of other existing HPLC methods and discuss them in comparison to the one described in the manuscript.
- Provide discussion of the presented ELISA kit results with other relevant publications.
- Check closely the reference list for compliance to the Guide to Authors.
- A thorough proofreading (English) is needed.
- ELISA kits are generally used as screening methods, yet the word “screening” is not present in the text. Also, the word “feed” is not present in the Keywords.
- For a comparative evaluation of commercial ELISA kits a full method validation should be presented. The protocols of each ELISA kit should be described in detail and the ELISA kits validation should include more performance characteristics (e.g. cross-reaction, specificity, stability, applicability, ruggedness, practicability etc. ). The preparation of the spiking working solutions should be stated.
- Likewise, more performance characteristics are needed for the HPLC-FLD method validation (see Commission Decision 2002/657/EC).
SPECIFIC COMMENTS
Line 21-22. “Both ELISA and HPLC were demonstrated to be appropriate methods for mycotoxin analysis”.
This generalization is a well-known fact. Consider rephrasing.
Line 22-23. “The availability and number of samples should be the primary determinants of the analytical process used”.
Unclear meaning.
Line 29. “Aspergillus, Fusarium, and Penicillium” and Line 36. “Aspergillus flavus and Aspergillus parasiticus”.
Use italics.
Line 31. “Aflatoxin”
Should be “aflatoxins”
Line 40. “they fluorescent”.
Use the verb not the adjective.
Line 41. “temperatures above 320°C without decomposing”.
Provide references.
Line 56. “Despite their flaws”.
Define these flaws.
Line 57. “HPLC is considered as the reference method for the qualitative and quantitative determination of mycotoxins [15]”.
Provide more references on HPLC or LC-MS analysis on mycotoxins, according to the latest developments.
Line 85. “spiking of milk samples”.
MILK????
Line 92. “Feed samples were obtained from individual feed selling businesses from various regions of Greece”. BUT in Line 285 “The samples were collected from individual farms in northern Greece”.
Correct accordingly.
Line 96-97. “high pressure liquid chromatography (HPLC)” BUT in Line 11-12 and Line 51“high-performance liquid chromatography (HPLC)”
Correct accordingly.
Line 102. “to symmetrically cover the analysis area at 50-150%”.
Rephrase for accurate meaning.
Line 103. maximum permissible legal limit (MRL).
Define MRL.
Line 110-112. “Prior to analysis, the extract was subjected to pH measurements for AgraQuant 1/20 and BioShield ES Total AFs, while for 111 Ridascreen Total AFs 100 μl of the filtrate was diluted with 600 μl of distilled water”.
Why?
Line 120. “The concentrates were kept constant at -20°C”.
Correct sentence with “stored”.
Line 123-124. “to prevent aflatoxins inactivation”.
Consider “degradation”.
Line 129. “a 100 L loop”.
Correct L to μL.
Line 131-132. “a mycotoxin C18 5 um 4.6 x 250 mm analytical column”.
Delete “mycotoxin” and correct to 5 μm.
Line 138. “0.45 m membrane filters”
Correct m to μm.
Line 138-139 “Prior to usage, cellulose nitrate 0.45 m membrane filters from Sartorius Stedium Biotech GmbH (Goettingen, Germany) were filtered through a glass vacuum filtering system received from Alltech Associates (Deerfield, IL, USA)”.
Unclear meaning.
Line 144. “Fioroni 13mm x 0.2m microfilters”
Correct m to mm.
Line 146-147. 2.2.2. Chromatography.
Start new paragraph.
Line 146. ‘a 20-200 L micropipette”.
Correct L to μL
Line 1. “The initial volume ratio was 22:22:56”.
Explain “initial”. Allocate 22:22:56 to A, B, C.
Line 156. 40 mL of purified water BUT in Line 160-161 washed with 10 mL of filtered water BUT in Line 163 purified water.
Define water quality accordingly.
Line 168-169. “duplicate analysis of the standard solutions supplied in each commercial package”.
Why not triplicate (standard deviation)?
Line 174. “(axis Ψ)”.
Axis Y.
Line 177. cept axis (intercept).
Define in detail.
Line 179-180. “Only in the case of Ridascreen Total AFs, the concentration calculated from the reference curve in ng kg-1 was multiplied by a factor of 35 and then divided by 1000 (μg kg-1 )”.
Why?
Line 201. Table 2 must be improved.
Line 204. “when analyzing the spiked feed samples three times in the same day (within-day repeatability)”.
Why not 6 times?
Line 197. Table 1.
Concentrations are given in μg/kg in two columns except for the last one (ng/g).
Please explain.
Line 247. “spiked milk samples”.
MILK????
In Figures 3 and 4b there is inconsistency in the peaks identification and respective retention times.
Why???
Line 250. “a signal to noise ratio of 3 and 10, respectively”.
Correct to S:N ratio of 3:1 and 10:1.
Line 253. “spiked milk samples”.
MILK????
Line 271. “There were some variations detected”.
Explain in detail.
Line 279. The choice of analytical procedures is primarily determined by the availability and number of samples.
Explain the meaning of this sentence.
Line 287. “Table 4. Distribution of the feed samples by type”.
The Table (distribution?) can be omitted and the information presented in the text.
Line 294. “With certain exceptions, the HPLC method appeared to be congruent with the ELISA results”.
Please elaborate on “certain exceptions”.
Line 299-300. ‘This discrepancy with the ELISA approach is thought to be due to overestimation of the latter”.
Rephrase to correct the meaning of the sentence.
Line 309. “…The kit with slightly greater performance was chosen…”.
Define performance characteristics used for the selection.
Line 314. “…separated from the achieved complex substrate of animal…”.
Unclear meaning.
Line 315-317. “Furthermore, the photochemical derivatization which followed in the current study was found to be a reagentless, fast and easy procedure”.
Unclear meaning.
Line 317-318. “Both ELISA and HPLC were…..mycotoxin analysis”.
This is already known.
Author Response
"Please see the attachment."

Reviewer 2 Report
The study is based on feedstuffs but all references, MRL and legislation concern foodstuffs.
There are several typos concerning milk and not feedstuffs (lines 81, 85, 247, 253)
Is it water in the mixture of durum wheat and barley? Why the sample was frozen at -20°C?
In the line 103 is "i.e." not "ie"
Is not correct to spike the solution after you have added MeOH, the problem is a wrong final concentration (line 107)
The correct unit for the solutions is for L, not for kg (line 115)
Wrong units in the lines 129, 146, 163, 154.
There is typos in the lines 174 (axis ?) and 177 (cept axis?)
There isn't the figure 1
In the figure 3 the order of AFG1 and AFB2 is reversed
The Decision 657 (typos on line 242!) not concerned mycotoxins. See EU Reg 519/2014.
Author Response
"Please see the attachment."

Reviewer 3 Report
Manuscript focuses on the efficiency of three commercially available ELISA kits to accurately detect and quantify mycotoxins in various agricultural commodities. Comparison of ELISA results with the 'gold standard' shows that the three kits are very useful. The research will be of interest to farmers and the food/feed industry who require rapid methods to test their products and ensure products in the supply chain are safe and compliant with regulatory requirement.
Minor comments
Table 5 - Please write AFs and LODs in full and delete TOTAL from the table.
Author Response
"Please see the attachment."
